# The Influence of Plant Extract on the Phase Equilibrium of Structure I Gas Hydrate in a Simulated Offshore Environment

Virtue Urunwo Wachikwu-Elechi [1,*] , Sunday Sunday Ikiensikimama [1] and Joseph Atubokiki Ajienka [2]

[1] Shell—JV Aret Adams Professorial Chair in Petroleum Engineering, University of Port Harcourt, Port Harcourt 500102, Rivers State, Nigeria
[2] Emmanuel Egbogah Chair of Petroleum Engineering, University of Port Harcourt, Port Harcourt 500102, Rivers State, Nigeria
* Correspondence: elechivirtue@yahoo.com

**Abstract:** Gas hydrate inhibitors, especially those used in offshore environments, are chemicals. These chemicals are synthetic in nature and pose both technical and environmental risks. This study emphasizes the influence of a Plant Extract (PE) on the phase behavior and equilibrium of structure I (SI) gas hydrate and its inhibition efficiency. The PE was screened using a mini flow loop. From the pressure-temperature phase diagram, the various weight percentages of the PE were able to disrupt the thermodynamic equilibrium conditions of the water and gas molecules to lower temperatures and increase pressures, which caused a shift in the equilibrium curve to an unstable hydrate formation zone. The pressure versus time plot as well as the inhibition efficiency plots for the PE and Mono Ethylene Glycol (MEG) were evaluated. Overall, the inhibition efficiency of the PE was higher than that of MEG for 1 wt% (60.53%) and 2 wt% (55.26%) but had the same efficiency at 3 wt% (73.68%). The PE at 1 wt% had the greatest inhibition effect and adjudged the optimum weight percent with a well-regulated phase equilibrium curve. This shows that PE is a better gas hydrate inhibitor than MEG, which is toxic to both human and aquatic life; therefore, it is recommended for field trials.

**Keywords:** plant extract (PE); mono ethylene glycol (MEG); inhibition efficiency (IE); phase diagram; thermodynamic equilibrium



## 1. Introduction

Since its discovery by [1], gas hydrates have been a nuisance for oil and gas industry. Water molecules are the main structural framework of hydrate crystals. The interaction between water molecules and other molecules is characterized by Van der Waals forces, while for water molecules, hydrogen bonding is the main force underlying the interaction between them. The void spaces or cages in the crystalline lattice are occupied by gas molecules. Hydrate formation occurs in the presence of water; hydrate formers, such as methane, ethane, propane, and butane, as well as nitrogen; nonhydrocarbons, including carbon dioxide ($CO_2$) and hydrogen sulfide ($H_2S$); low temperatures (0–40 °C or 32–104 °F); and high pressures (>200 psi) [2]. Hydrates have also been said to form below 0 °C and at pressures of 200 psi. To obtain a stable crystal structure, a sufficient number of cages must be filled with hydrate formers.

The avoidance of hydrate formation conditions is very important to avoid problems such as choking on the flow strings, flow lines, surface, and other equipment. A decrease in the measured head pressure in the flow strings as well as the total blockage of the flow lines and surface equipment can also occur when hydrates form. If the gas hydrate plugs clump and travel at high velocities, they can cause equipment damage. Typically, there are three hydrate crystal structures. Structure I (sI) hydrates are formed by carbon dioxide, methane, ethane, and hydrogen sulfide. Structure II (sII) hydrates are peculiar to the oil and gas industry and are formed by propane, isobutane, and nitrogen. Structure H (sH) hydrates are not common and are formed by cyclo-alkanes, paraffin, and pentane [3,4].

According to [5], gas hydrate structures are stabilized by guest molecules that are trapped in the lattice structure of gas hydrates, and without the guest molecules, the lattice structure will collapse. Hydrate structures are composed of 15% gas (also known as guest molecules) and 85% water (commonly called the host molecule) [6]. A study by Rajnauth et al. (2010) reports that the gas hydrate formation pressure and temperature are dependent on the composition of the guest molecules. Gas hydrate formation can occur during oil and gas production; during drilling, especially in control lines; during improved oil recovery using carbon dioxide gas; and in other systems wherever the circumstances of its formation are satisfied [7–9].

To prevent hydrate formation, various methods have been applied. They include fluid separation to remove hydrate formers via heat application, pressure control, mechanical scrapping or pigging, thermal insulation, electrical heating, and chemical injection, which is the most common and effective method, especially in areas such as deep offshore environments where accessibility is a problem [10–13]. These chemicals are termed inhibitors and are subdivided into:

**(1) Thermodynamic Hydrate Inhibitors (THIs):** They modify the chemical ability of the hydrate or aqueous stage [14]. They change the balance conditions of the gas hydrate curve to pressures and temperatures that do not favor the formation of hydrates. This is mostly required at large concentrations of 10–50 wt% for water cuts [14]. Methanol (MeOH) and Ethylene Glycol (EG) are commonly used. Thermodynamic hydrate inhibitors face many drawbacks even though they are still currently used in the field. Their disadvantages include the high cost of operation as a result of transportation costs, the storage cost, the injection and pumping quantities required, and regeneration units in the case of glycols [15,16].

**(2) Low Dosage Hydrate Inhibitors (LDHIs)**: These are so termed because of the volume needed to inhibit gas hydrates. They are used in doses as low as 0.01–1 wt% for water cuts. They are subdivided into Kinetic Hydrate Inhibitors (KHIs) and Anti-Agglomerants (AAs).

**(a) Kinetic Hydrate Inhibitors (KHIs):** They are polymeric water-soluble compounds that slow down hydrate formation by causing an increase in the energy required for hydrate formation through structural distortion [17]. This is achieved by allowing hydrate crystals to grow between and around the polymer strands through gas diffusion, resulting in the blockage of water [18]. Variations in the water cuts favor their actions, but they are sensitive to brine salinity [19]. Examples include polyvinylpyrrolidone and vinyl caprolactam, which were first discovered by the Colorado School of Mines in the early 1990s [20].

**(b) Anti-Agglomerants (AAs):** Anti-agglomerants act as surface active agents (i.e., surfactants) and prevent hydrates from sticking together and clumping. The hydrate still forms, but the crystals do not plug and can be transported through pipelines due to the small crystal size. They work only in the presence of a liquid hydrocarbon phase, i.e., crude oil or condensates. An example is alkyl aromatic sulphonates.

Unlike THIs, such as methanol or glycols, hydrate formation is not prevented by the use of Low Dosage Hydrate Inhibitors (LDHIs), because a hydrate phase boundary shift does not occur. The elimination of hydrates is impossible once they form because the conditions under which the operation takes place cannot be changed. Subsequently, Thermodynamic Hydrate Inhibitors are vital when a well is about to be kicked off or shut in. Even though KHIs help to reduce operating and capital costs, their large-scale field application is still problematic, because they are not environmentally friendly; that is, they degrade poorly when exposed to the environment, as posited by researchers such as Jensen et al. and Kelland et al. In light of this and to address the issues of environmental footprints, research is moved towards the formulation of inhibitors that are environmentally friendly and biodegradable. These inhibitors are termed green inhibitors.

**(3) Green Inhibitors (GIs):** GIs are so termed because they are non-pollutants, biodegradable, and environmentally friendly. They include Anti-freeze Proteins, AFPs [21–24], Natural and Biodegradable polymers, NBPs [25,26], and Ionic Liquids, ILs [27] (particularly liquids that are imidazolium-based) have been tested for the mitigation of gas hydrate formation.

The use of various hydrate inhibitors has been investigated by various researchers. The first successful use of KHI was recorded by [28], with its successful application that replaced the use of THIs (methanol and glycol) in a wet gas pipeline connecting the Hyde-West sole gas field to an onshore Easting terminal in Southern North Sea (UK). About 5500 parts per million (ppm) of the KHI was required with just a storage tank and metering pumps which cut down logistic costs tremendously. KHIs reduce operating costs (OPEX). According to [17], its use instead of methanol in a deep offshore pipeline in the Gulf of Mexico (GoM), saved costs of between one hundred and twenty-five thousand and one hundred and forty thousand dollars per month (125,000 USD/month and 140,000 USD/month). The first commercial application of a kinetic Hydrate Inhibitor (KHI) developed by Exxon in a crude oil system was presented by [29], after laboratory testing in Exxon's 4-inch diameter hydrate flow loop at pressures up to 1800 psig, the KHI was field tested in a 2-inch diameter, 1.5 mi gas flow line in Alberta, Canada between 1996 and 1997. The conclusion was that in a crude-oil system, the KHI developed by Exxon was effective.

In 2002, [30] carried out a comprehensive laboratory evaluation after which he selected an anti-agglomerant (AA), a low-dosage hydrate inhibitor (LDHI) for field testing in deep water, in a Gulf of Mexico (GoM) oil well. He had an earlier insight that LDHIs have helped in demulsifying some black oil emulsions. The selected LDHI did not affect the overboard water quality and did not cause any emulsion problems which consequently caused the Basic Sediment and Water (BS and W) counts to remain low. In a study carried out by [31], induction time was observed to be extended with the use of inhibitors that had an order of magnitude to only the inhibitor when determining the gas uptake, induction times, and temperature. They also noted that although Polyethylene oxide (PEO) was not an inhibitor, it enhanced the performance of the kinetic inhibitor when added to the KHI solution. Mono Ethylene Glycol (MEG) and methanol (MeOH) were compared as gas hydrate inhibitors during gas expansion by [32] using CSM Gem software to generate inhibition performance and hydrate formation curves. It was observed that methanol performed better than Mono Ethylene Glycol at 10 wt% and was able to inhibit hydrate formation when expanded to 3338 psi but even at a gas expansion beyond 3750 psi, 10 wt% of Mono ethylene glycol failed to inhibit gas hydrate formation.

The use of naturally occurring locally available compounds as gas hydrate inhibitors is scarcely seen in the literature on gas hydrate inhibition. A study by [33], using methanol, a commercial gas hydrate inhibitor as a benchmark to evaluate the effectiveness of the local reagent (LR) was carried out. The pressure profiles as well as inhibition plots showed that the LR performed better in all but 1 wt%. They opined that the presence of bioactive compounds aided in its effectiveness. A study on the performance of a local surfactant, Surf. X derived from plant material was investigated by [34]. The effectiveness of Surf. X was compared to that of N-Vinyl Caprolactam (N-VCap) in a locally fabricated laboratory flow loop. Various analyses showed that the Surf. X had a better performance in all but 0.04 wt%. They suggested its development as a gas hydrate inhibitor. In the study done by [35], the efficiency of a local surfactant used as KHI was compared to methanol a commercial THI. The study showed that even in small quantities when compared to MeOH that was used in higher weight percentages, the Locally Sourced Surfactant (LSS) performed better than the MeOH in all weight percentages considered. Indeed, refs. [36,37] also studied the use of local materials and agro-waste as gas hydrate inhibitors and [38] reported the use of plant extract (PE) as a gas hydrate inhibitor. The experiment was conducted in a simulated offshore environment of a mini flow loop apparatus, 39.4 inches long having an internal diameter of 0.5-inches encased in a 4-inch Poly Vinyl Chloride (PVC) pipe skid mounted on a metal framework. The local inhibitor was reported to contain bioactive compounds such as alkaloids, saponins, tannins, and flavonoids and was used in varying weight percentages (1–3 wt%) with water. Plots of pressure versus time, temperature versus time, differential pressure and pressure, and temperature versus time for both inhibited and uninhibited scenarios were used to evaluate the performance of the plant extract (PE). It was concluded that for 1 and 2 wt%, a better inhibitory effect was observed for the Plant Extract (PE) as

compared to the same weight percentage of Mono Ethylene Glycol (MEG). For 3 wt%, both Plant Extract and Mono Ethylene Glycol showed a close match in inhibitory capacity. The PE is recommended for field trials since it is eco-friendly and biodegradable. In the study by Elechi et al. [38], the ability of the Plant Extract (PE) a local inhibitor to inhibit hydrate was shown without particular attention to the inhibition efficiency. The need for more research on the use of locally sourced materials in place of synthetic toxic and expensive inhibitors is apt. This study, therefore, considers the inhibition efficiency of the Plant Extract (PE) and the optimum weight percentage of the plant extract (PE) based on its inhibition efficiency.

## 2. Materials and Methods

The main experimental apparatus used for this study is a 39.4 m closed locally fabricated laboratory mini flow loop made of 316 stainless steel with an internal diameter of 0.0127 m (0.5-inch enclosed in fiber wool insulated 0.1016 m Polyvinylchloride (PVC) pipe skid mounted on a metal framework. It has a control panel, three pumps between 0.5–1 hp (for agitation of fluid and movement of fluid in and around the loop and stainless steel pipe), pressure (P1–P6), and temperature gauges (T1–T3), an Orifice and Valves (V1–V7, for controlling the gas inflow and for opening and closing various parts of the system), a flow meter (for seeing mixing vessel (for mixing varying concentrations of inhibitors and water), a refrigerating unit (mimics the offshore environment) and a compressed natural gas (CNG) cylinder (containing hydrate formers) as shown in Figure 1, Scheme 1, and Table 1.

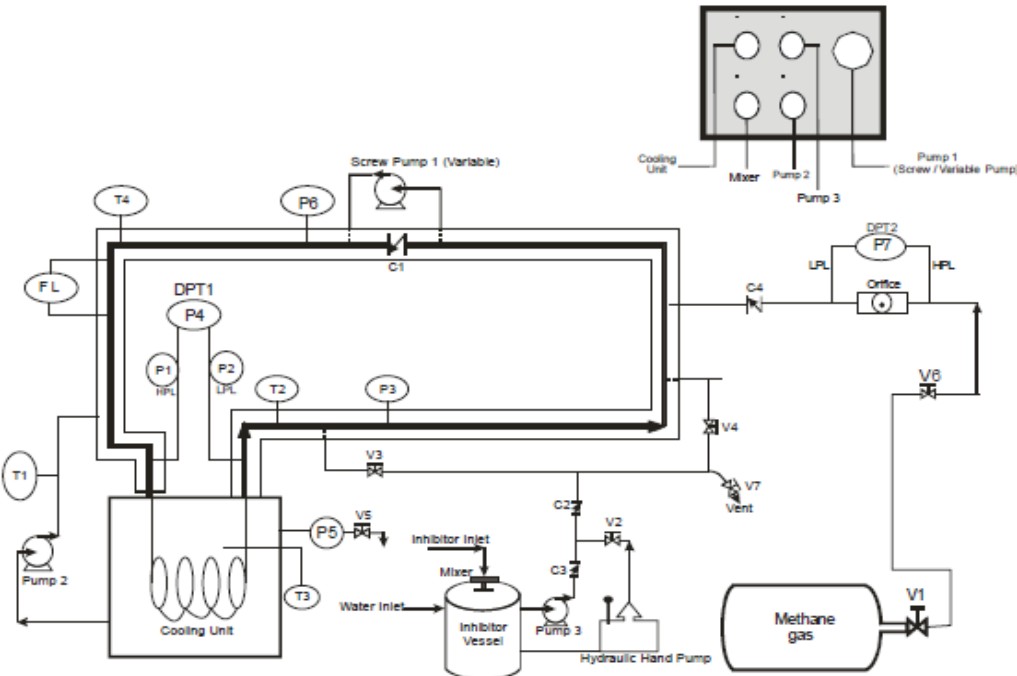

**Figure 1.** Process Flow Diagram (PFD) of laboratory mini flow loop [39].

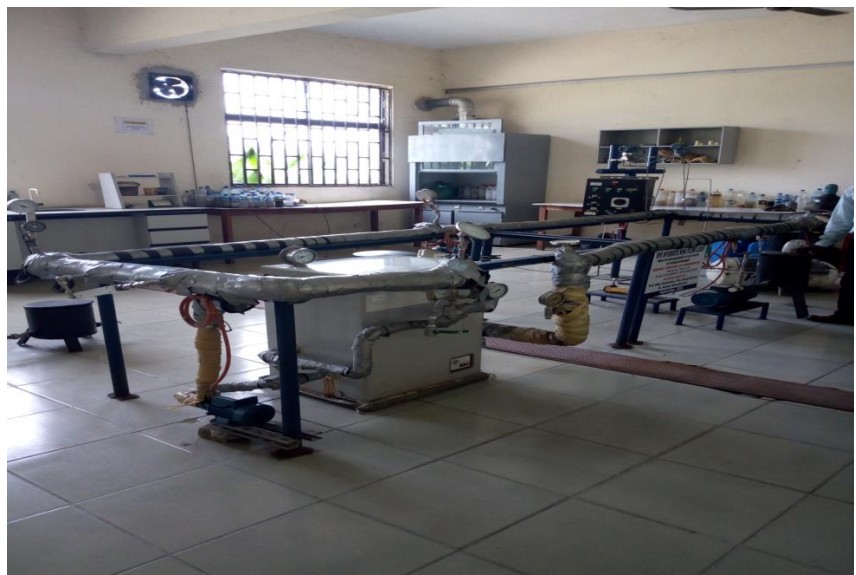

**Scheme 1.** Laboratory Mini Flow Loop for Hydrate Study.

**Table 1.** Composition of Compressed Natural Gas Used.

| Components | Molecular Weight (Mw) | Mole Fraction (%) |
|---|---|---|
| Methane, $CH_4$ | 16 | 98.44 |
| Carbondioxide, $CO_2$ | 44 | 1.56 |
| Hydrogen sulfide ($H_2S$) | 34.08 | - |
| Total Sulfur | 32.065 | - |
| Oxygen ($O_2$) | 16 | - |

*2.1. Assumption/Limitations of Research Work*

1. The Mini flow loop used operates within a Loop pressure of 3500 psi and temperature between 0 and 50 °C.
2. The maximum allowable pressure for the loop was 150 psi because above this pressure, the screw pump failed because the pump used was for single-phase liquid flow so the quantity of pumped gas had a limit.
3. The system operates as a constant volume batch process therefore the amount of gas used up is reflected in the pressure of the system at the end of the experiment.
4. Rapid temperature increases and excessive decreases in pressure were an indication of hydrate formation in the system. This is so because hydrate formation is an exothermic reaction indicated by temperature increase. The pressure decrease is due to a reduction in the number of gas molecules in the system
5. The system is made of 316 stainless steel pipes that are insulated inside a 4-inch PVC pipe with cold water circulated constantly to cool the stainless-steel pipe, mimicking the offshore environment.
6. The system studies gas hydrate formation in a gas-dominated two-phase flow system and predominantly studies how pressure affects gas hydrate formation in a gas-dominated 2-phase system.
7. About 1 m of the 0.5-inch internal diameter pipe is spiraled and exposed inside the refrigerator (cooling unit) to increase the retention time of the hydrate-forming fluid in the coldest part where gas hydrate is likely to form.

Materials used include tap water, ice blocks, and compressed natural gas with a specific gravity of 0.5 composed mainly of 98.44-mole percent methane and 1.56-mole percent of carbon dioxide.

The Plant Extract (PE) was obtained from Arecaceae Family Exudates and contained equal amounts of saturated and unsaturated fatty acids. The unsaturated fatty acids contain oleic acid, palmitic acid, and stearic acid as the main active components. The acid contains a carbonyl group and a double bond in them [40]. They contain large molecular structures, double bonds, and reactive groups that contributed to the ability to cover a large surface area of the metals [41]. It is reported to contain biochemical compounds such as sugar, proteins, amino acids, alcohol, and minerals [42].

From the phytochemical screening done, the constituent of the plant extract (PE) is flavonoids (1.6%), Saponins (3.1%), alkaloids (10.6%), and tannins (6.5%). Tannins are high molecular weight polyphenolics found to be most abundant in nature and concentrated in the leaf's tissues, epidermis, bark layers, flowers, and fruits of plants as condensed tannins. They form reversible and irreversible complexes in an aqueous medium because they are heterogenous polyphenolic compounds with high molecular weight. They prevent oxidation. Flavonoids are polyphenolics that are known to fight inflammation and prevent clumping. Heterocyclic nitrogen atoms which are naturally occurring and produced by a vast number of living cells called Alkaloids are also contained in the local reagent and they also prevent coagulation. Saponins are surface active agents that form foams as well as bubbles which give good stability and prevent agglomeration. They act against reactive oxygens that cause oxidation [43]. The ability of the local reagent to perform as a gas hydrate inhibitor is aided by the presence of these bioactive compounds.

*2.2. Procedure*

The locally fabricated mini flow loop functions as a closed system that has constraining conditions, having a constant volume and amount of components in the system. The phase changes as well as component exchange in the system were defined by the existing conditions of the flow loop defined such that pressure changes are temperature and phase-dependent. This means that as the pressure of the loop varied, the volume of gas in the loop also varied as an indication that more gas was used for hydrate formation. For this system, plugging of the sample point valve, increase in loop temperature, increase in differential pressure, rapid decrease in loop pressure, and cloudy or milky color of effluents from the sample point were all indications of gas hydrate formation. The method used in the experiment had to do with monitoring parameters such as pressure and temperature of the system, taking into cognizance the exothermic reaction and depletion of gas molecules during hydrate formation. This is in line with the work of [44].

Figure 2 is the flow chart for the experimental run. Before the commencement of the experimental run, water is poured into the mixing vessel. Pump 3 is turned on to let about 435 mL of water into the 0.5-inch diameter tubing till a pressure of 25 psi is attained and the Pump is turned off. Valves 5 or 7 are used as a vent to let out the water in the inner line. The aim is to ensure that debris in the inner line is removed, hence, the uptake and venting of water into and from the system is done repeatedly until the aim is achieved. For the hydrate formation experiment, about 2660 mL of water is poured into the mixing/inhibitor vessel, and Pump 3 is turned on to build up the pressure of the system to 25 psi initially. This process allows about 435 mL to enter into the inner line after which the pump is turned off. The pressure of the system is then built up to 150 psi by turning on Valve 1 and the Orifice which allows gas to flow from the CNG cylinder into the inner line. When this pressure is attained, the valve and Orifice are turned off, and Pump 2 is turned on to let the circulation of water from the refrigerator into the PVC pipe and back to the refrigerator. To quicken temperature reduction to hydrate formation temperature, ice blocks are added to the refrigerator. To cause agitation and move the fluid around the inner line, the screw pump is turned on and set at 250 V. Temperature and pressure readings are taken every two minutes for two hours (120 min). When the temperature of the loop begins to raise drastically (as a result of heat being released) or the pressure of the loop decreases rapidly (when more moles of gas are used up to form hydrate) or there is an increase in differential pressure (as a result of restrictions due to deposition in the inner line which reduces the

internal diameter) the hydrate is said to form in the uninhibited experiment. The hydrate inhibition experiments are also conducted in the same vein but the inhibitor vessel does not contain just water, but also the inhibitors in their varying weight percentages to water cut and then the experiment proceeded as described above for 120 min.

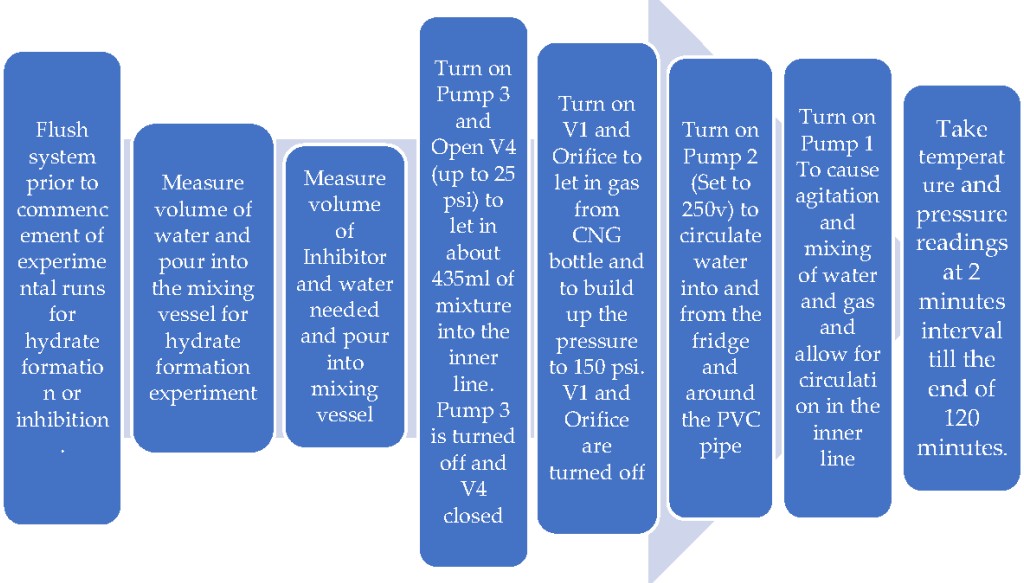

**Figure 2.** Flow Chart of the Experimental Procedure/Run.

## 3. Results and Discussion

Plots of pressure versus time were done to evaluate the performance of the local inhibitor alongside the conventional inhibitor. The performances of both are based solely on the performance of the systems without inhibitors which is used as the control experiment. Furthermore, the performance of the Plant Extract (PE) is based on the performance of Mono Ethylene Glycol (MEG). This means that the performance of MEG was a yardstick/benchmark for assessing the performance of the Plant Extract (PE). In the same vein temperature versus time plots were also plotted to evaluate the inhibitors' performance as stated above. The inhibitors were evaluated on three different weight percentages (1 wt%, 2 wt%, and 3 wt%)

Figure 3 shows the experimental run with gas and water and no inhibitor (Pressure versus Time plot). For the first twenty (20) min, there is a reduction in pressure from a base pressure of 150 psi to 115 psi. This initial rapid drop in pressure of the system is due to the dissolution of gas in the water inside the 0.5-inch ID inner line encased inside the PVC pipe. This stage is the initial rapid drop in the pressure stage of hydrate formation. The pressure further reduced to 45 psi after 75 min into the experiment and reduced steadily to 44 psi, 42 psi, and 40 psi in about 100 min of the experiment. This constant decline is a result of hydrate crystal nucleation or induction. A further reduction to 36 psi which was maintained till the end of 120 min is a result of hydrate crystal growth. This agrees with the literature by Bishnoi and Natarajan [45] and Jensen et al. [46]. Temperature decreased from 30 °C to 20 °C in 20 min and in the next 20 min, it increased to 27.5 °C and then 29.5 °C after an hour which was maintained till the end of the experiment. Hydrate formation is confirmed in this system due to the rapid decline in pressure and rise in temperature. At the hydrate formation zone, rapid crystallization occurs due to an increase in nucleation (hydrate nucleation is due to adsorption and clustering at the interface of the vapor/gas side where gas molecules move to the interface and adsorb in the aqueous/liquid phase) which begins with a sudden pressure drop and continues until hydrate growth is completed as a result of partial or complete cavities [47].

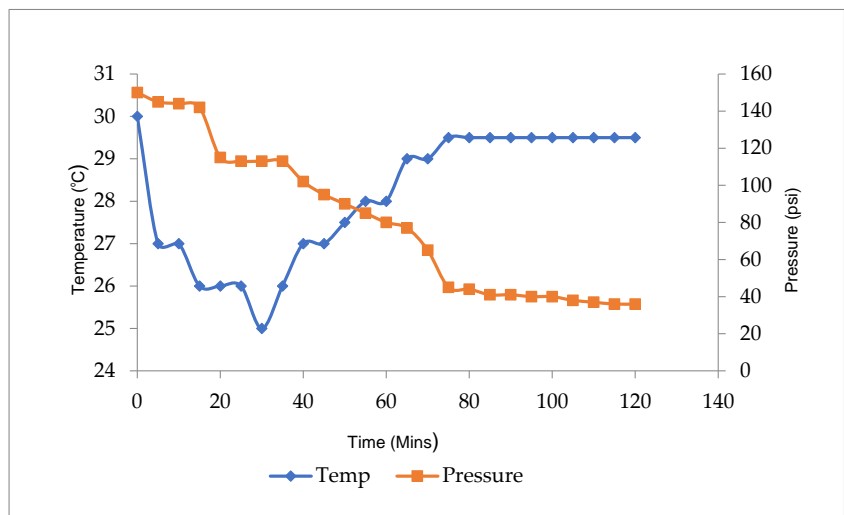

**Figure 3.** Pressure and Temperature versus Time for Water and Gas System.

Figure 4 shows the plot of pressure and temperature versus time for 1, 2, and 3 wt% Plant Extract (PE). For 1 wt% of Plant Extract (PE), the change in pressure was 10 psi in the first 20 min as compared to the uninhibited experiment with a change in pressure drop of 35 psi. Pressure gradually reduced to 134 psi after an hour and was maintained for the next 15 min after which it reduced to 132 psi and was maintained for the remainder of the experiment. Temperature decreased from 28 °C to 11 °C after 60 min and further to 9 °C in the next 15 min and maintained till the end of the experiment. There was no rise in temperature or rapid drop in pressure in the system as noticed in the system using water and gas only without inhibitor as seen in Figure 4. For the plot of pressure and temperature versus time for 2 wt% Plant Extract (PE), pressure decreased from 150 psi to 120 psi giving a change in pressure of 30 psi in the first 20 min, for 2 wt% of PE. It further decreased to 106 psi after 60 min and then 105 psi and was maintained till the end of the experiment. The temperature of the system reduced from 30.5 °C to 19.5 °C in 20 min and then to 6.5 °C after an hour, this value was maintained for 25 min and finally dropped to 6 °C till the end of the experiment as seen in Figure 4.

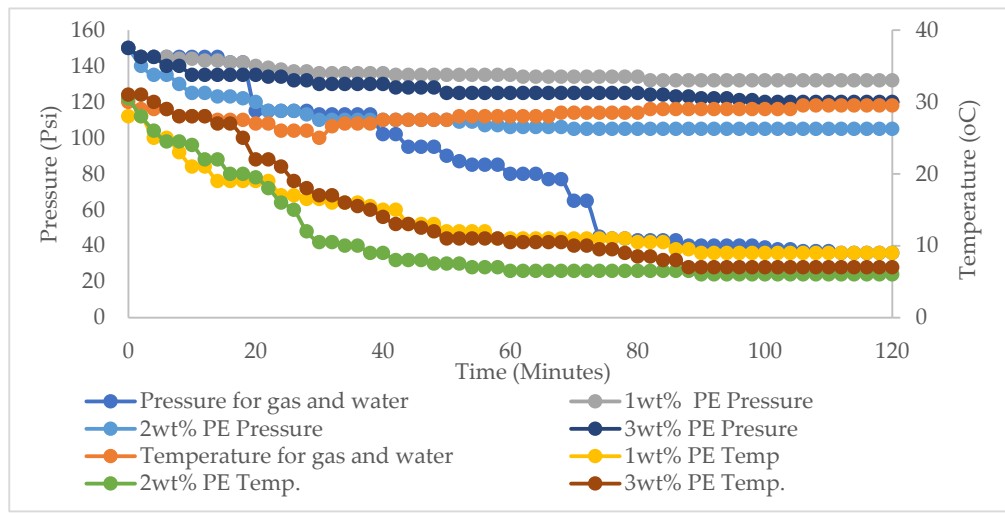

**Figure 4.** Pressure and Temperature versus Time for 1, 2 and 3 wt% Plant Extract (PE).

For 3 wt%, the initial pressure drop was from 150 psi to 135 psi in the first 20 min giving a change in pressure of 15 psi. It further reduced to 125 psi after 60 min, was maintained for 25 min, and then dropped to 120 psi which it maintained till the end of the

experiment (120 min). Temperature declined from 31 °C to 22 °C in the first 20 min and then to 10.5 °C after 60 min after which it reduced to 7 °C in 15 min and was maintained till the end of the experiment.

Since the pressure and temperature for 1 wt% of the PE after two hours was 132 psi and 9 °C while that of 2 wt% was 105 psi and 6 °C and that of 3 wt% 120 psi and 7 °C, the best weight percentage for hydrate inhibition using this inhibitor, is 1 wt%, further increment had no significant effect on hydrate inhibition. In all the weight percentages, the Plant Extract was able to prevent hydrates given the fact that there was no unusual temperature and pressure decline in the systems using the Plant Extract. The extract was able to interfere with the hydrogen bonding of water molecules (it competed more with water molecules in terms of hydrogen bonding) which helps to shift the hydrate-liquid vapor equilibrium curve to higher pressures and lower temperatures thereby creating a change in thermodynamic properties of hydrate formation at the given condition making hydrate formation thermodynamically less likely [47].

### 3.1. Hydrate Equilibrium Pressure-Temperature Plots for 1, 2, and 3 wt% PE and Uninhibited Experiment (Water and Gas)

The pressure/temperature relationship that forms and causes hydrate dissociation is defined by the hydrate formation and dissociation curve. It defines the temperature and pressure envelope where the subsea hydrocarbon system has to operate at a steady state and transient conditions to avoid the possibility of gas hydrate formation [48]. Figure 5 is the pressure versus temperature plot for 1, 2, and 3 wt% PE and Uninhibited Experiment (Water and Gas). The various weight percentages of the PE were able to disrupt the thermodynamic equilibrium conditions of the water and gas molecules to lower temperatures and higher pressures which caused a shift in the equilibrium curve to the left allowing for an unstable hydrate formation zone. In fact, 1 wt% shifted the curve to the left better having a pressure of 132 psi at the end of the experiment (Figure 5). The performance at this weight percentage could be attributed to better interaction with the water molecules which did not allow for the formation of gas hydrates. Indeed, 2 wt% and 3 wt% did not perform as well as 1 wt% even though the more the inhibitor content, the higher the temperature drops and the higher the pressure [49]. This could be due to lesser activity or could be a case of over-inhibition leading to a reduction in efficiency. This also aligns with a study based on ionic liquids where it was opined that concentrations greater than 1 wt% did not portend a significant advantage [20].

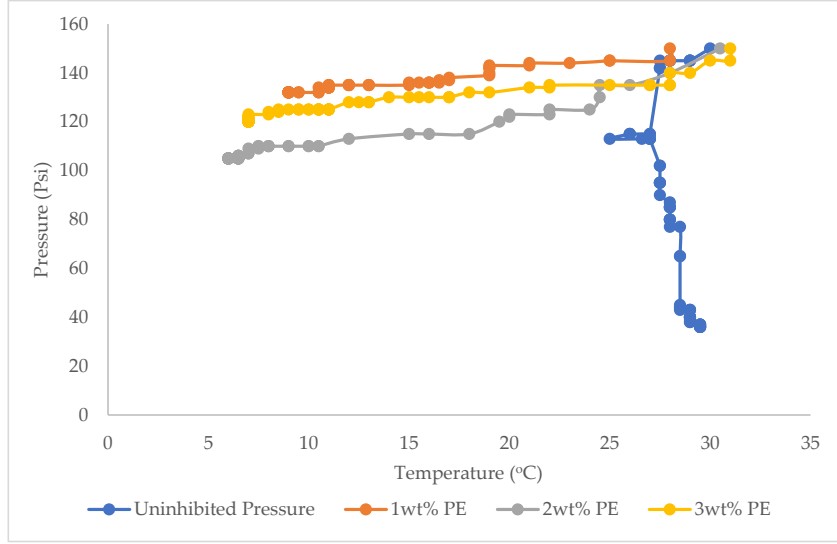

**Figure 5.** Pressure versus Temperature plot for 1, 2 and 3 wt% PE and Uninhibited Experiment.

The uncharged clusters in the inhibitor molecules produce an interactive force with water molecules and disrupt the liquid water molecule cages generated by the hydrogen bond. This means that the water molecules need to overcome this interactive force to form cages, consequently, gas hydrate will need additional energy to alter the hydrate formation pressure and temperature condition from actual operating conditions [49].

### 3.2. Comparison of the Inhibitory Capacities of Plant Extract (PE) and Mono Ethylene Glycol (MEG)

From the assumptions/limitations earlier stated, the system studies gas hydrate formation in a gas-dominated two-phase flow system and predominantly studies how pressure affects gas hydrate formation in a gas-dominated 2-phase system. In light of this, the effectiveness of the inhibitors was evaluated based on the pressure variations in the system at the end of the experiment. Again, the system operates as a constant volume batch process therefore amount of gas used up is reflected in the pressure of the system at the end of the experiment which is also indicative of the performance of the inhibitor.

A comparison of Plant Extract (PE) with Mono Ethylene Glycol (MEG) in terms of pressure versus time, is shown in Figure 6. For 1 wt%, the overall pressure of the system with MEG was reduced to 105 psi while 2 wt% gave a reading to be 99 psi and the value of 3 wt% to be 120 psi. Pressure drops for the system using PE for 1, 2, and 3 wt% were 132 psi, 105 psi, and 120 psi. PE did better in inhibition in 1 and 2 wt% but had the same value with MEG for 3 wt% at the end of the experiment. PE is shown to be a better inhibitor than MEG.

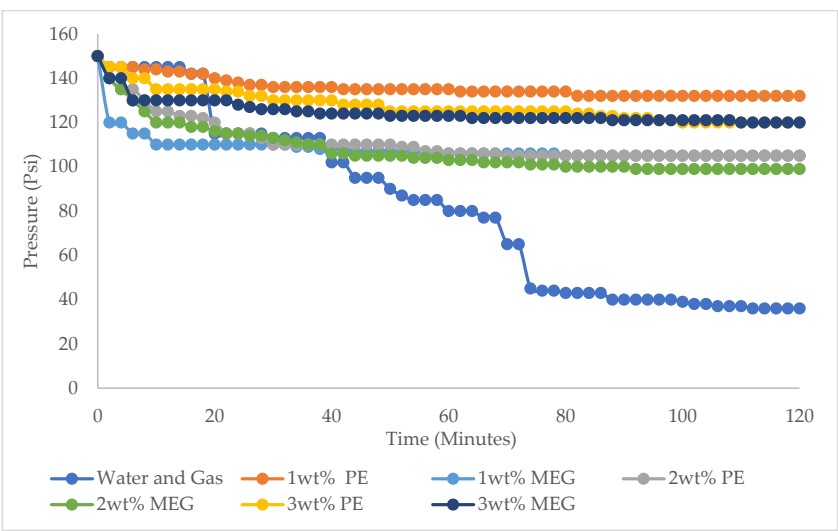

**Figure 6.** Pressure versus Time for 1, 2 and 3 wt% of Plant Extract (PE) and Mono Ethylene Glycol (MEG).

### 3.3. Hydrate Equilibrium Pressure-Temperature Plots for 1, 2, and 3 wt% PE, MEG and Uninhibited Experiment (Water and Gas)

Figures 7–9 are the pressure versus temperature plots for 1, 2, and 3 wt% PE, MEG, and the uninhibited Experiment. For the Uninhibited experiment (water and gas), the pressure and temperature had an inverse relationship as the experiment progressed. As the pressure decreased initially, the temperature also decreased with time. This aligns with [50] that stated that pressure decreases correspondingly as temperature decreases at a constant rate for a closed system. As the residence time of the fluid in the system increased (above 30 min at a pressure of 113 psi and temperature of 25 °C), the induction time was shortened and hydrates began to crystalize. This was seen in the pressure-temperature profile as the temperature began to raise till the end of the experiment to 29.5 °C. The pressure reduced drastically to 36 psi at the end of the experiment. This led to the shifting of the pressure-temperature curve to the right giving more room for hydrate formation to the left of the curve. To the right of the dissociation curve is the region in which hydrates

do not form so operating in this region is safe from hydrate plugs. To the left of the hydrate curve is the region where hydrates are thermodynamically stable and have the potential to form [51].

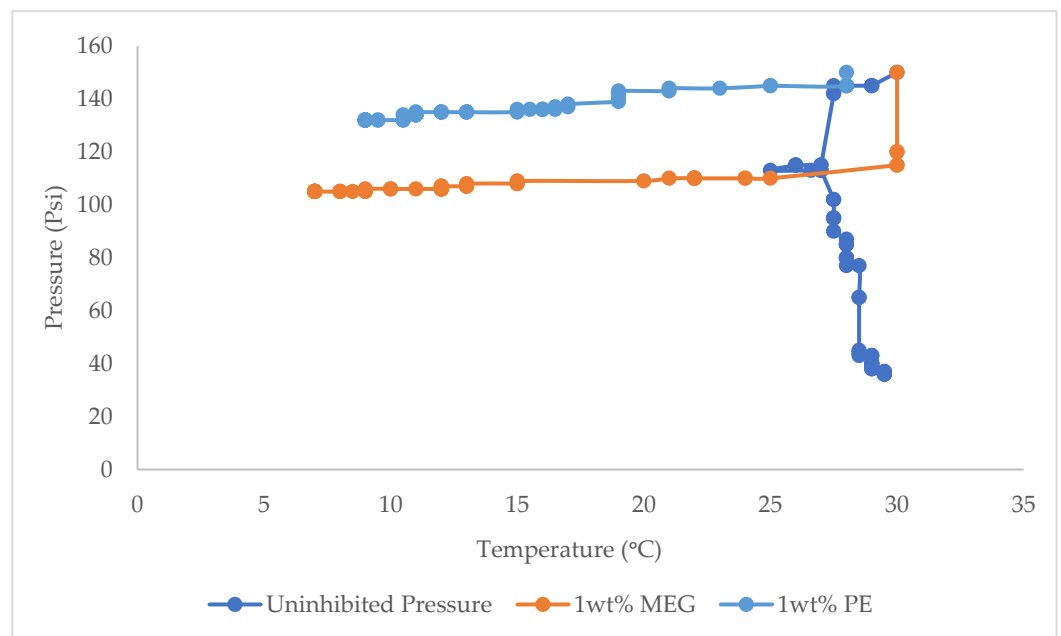

**Figure 7.** Pressure versus Temperature for 1 wt% Plant Extract (PE), Mono Ethylene Glycol (MEG), and Uninhibited Experiment (water and Gas).

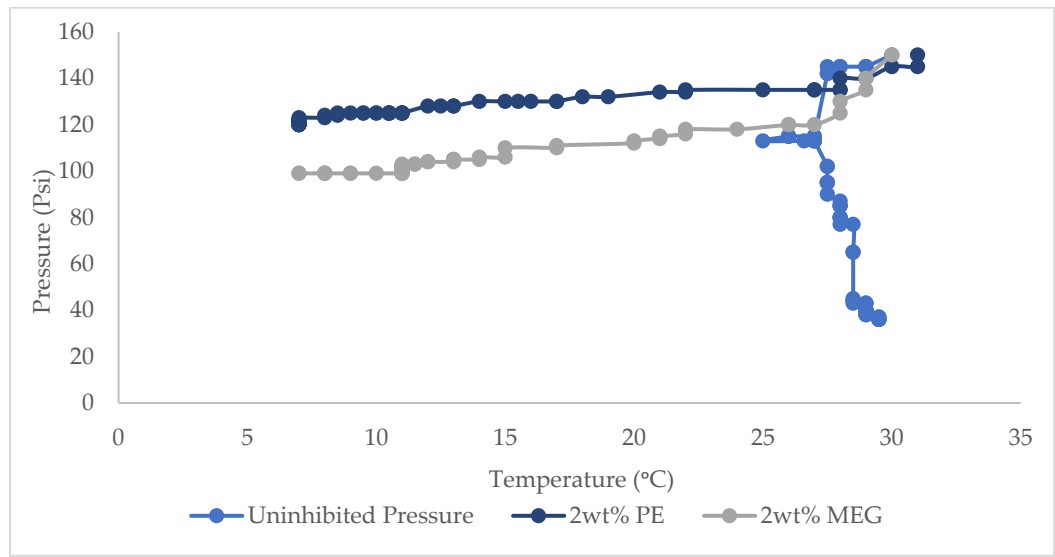

**Figure 8.** Pressure versus Temperature for 2 wt% Plant Extract (PE), Mono Ethylene Glycol (MEG) and Uninhibited Experiment (water and Gas).

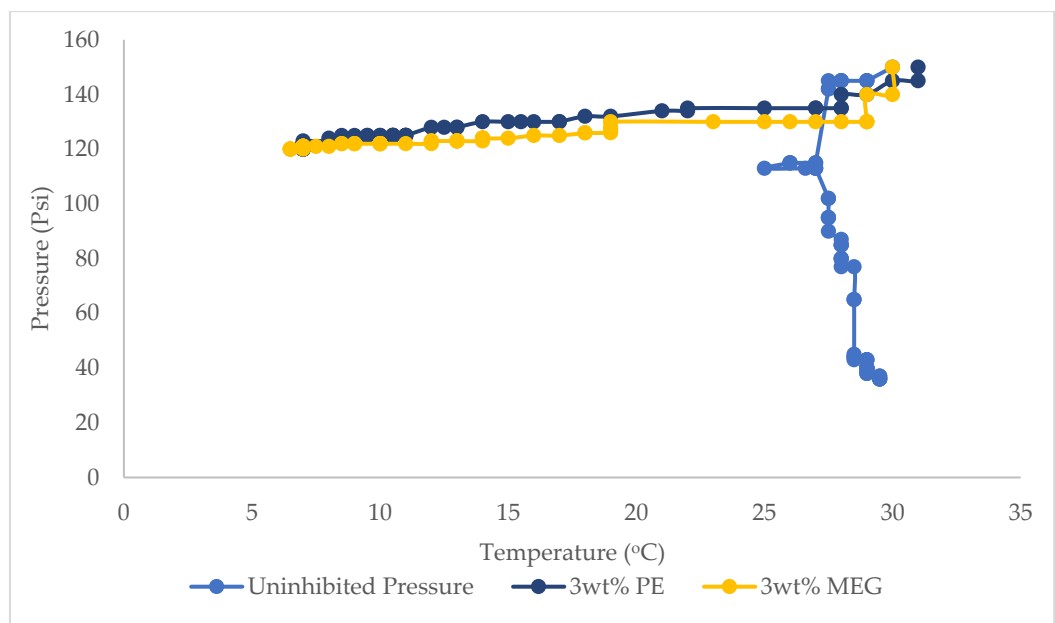

**Figure 9.** Pressure versus Temperature for 3 wt% Plant Extract (PE), Mono Ethylene Glycol (MEG), and Uninhibited Experiment (water and Gas).

Thermodynamic hydrate inhibitors delay gas hydrate formation by reducing hydrate formation temperature via a change in the chemical potential of water [49]. It shifts the hydrate equilibrium curve to the left to lower hydrate equilibrium temperature in such a way that the system is kept out of the hydrate formation zone [52] For 1 wt% PE, the pressure-temperature shift to the left was from 150 psi at a temperature of 30 °C to 132 psi and a temperature of 9 °C. For the same weight percentage for MEG, the shift was the same but with lesser pressure (105 psi and 7 °C). For 2 wt% PE and MEG, the values were 105 psi at 6 °C and 99 psi at 7 °C. 3 wt% pressure values at the end of 120 min were the same for PE and MEG at 7 and 6.5 °C, respectively. For the inhibited experiment, 1, 2, and 3 wt% of PE and MEG caused a shift in the phase envelope to the left to lower temperatures and allowed for more room to the right so that hydrate formation is not stable, but PE had a more stable hydrate free zone than MEG. The PE probably had more hydrogen bonding affinity as a result of the presence of more OH groups. The more the equilibrium line tilts to the left, the larger or longer the safe area or zone which is the condition that prevents gas hydrate formation [53]. The performance of MEG could be attributed to its high viscosity and lesser bioactive compounds/functional groups (having just 2-OH groups). The inhibitor molecules competed with the water molecule and changed the thermodynamic equilibrium of the water and hydrocarbon molecule (i.e., changing the chemical potential of hydration), and prevented hydrate formation by moving the phase equilibrium curves to lower temperatures and higher pressures which makes the hydrate to become unstable, decompose, and easily separate [47].

There was no drastic pressure decrease as shown in the uninhibited experiment where pressure decreased to 36 psi as a result of the rapid dissolution of gas in water leading to more gas usage in hydrate formation [24]. This caused the P-T curve to tilt very much to the right making hydrate formation very stable.

*3.4. Initial and Final Pressure versus Time for 1, 2, and 3 wt% of PE, MEG, and Uninhibited Experiment*

Table 2 shows the initial and final pressures and changes in pressure for the uninhibited system, 1, 2, and 3 wt% of PE and MEG. For the system with Gas and water only, the Pressure decreased from 150 psi to 36 psi in 120 min giving a change in pressure of 114 psi. The Plant Extract (PE) gave a change in pressure for 1 wt% at 18 psi, 2 wt% at 45 psi, and 3 wt% at 30 psi. For the system with Mono Ethylene Glycol (MEG) pressure changes for

1, 2, and 3 wt% of the inhibitor were given as 45 psi, 51 psi, and 30 psi respectively. This shows that PE did better in inhibiting hydrates than MEG, especially in 1 wt%. The plot comparing PE and MEG is shown in Figure 5 and it goes a long way to ascertain PE is a better inhibitor than MEG.

**Table 2.** Initial and Final pressure values and change in pressure for Gas and Water, 1 wt%, 2 wt%, and 3 wt% of Plant Extract (PE) and Mono Ethylene Glycol (MEG).

| | 1 wt% | Δp (psi) | 2 wt%(psi) | Δp (psi) | 3 wt%(psi) | Δp (psi) |
|---|---|---|---|---|---|---|
| **Gas and water** | 150 − 36 | 114 | 150 − 36 | 114 | 150 − 36 | 114 |
| **Plant Extract (PE)** | 150 − 132 | 18 | 150 − 105 | 45 | 150 − 120 | 30 |
| **Mono Ethylene Glycol (MEG)** | 150 − 105 | 45 | 150 − 99 | 51 | 150 − 120 | 30 |

*3.5. Inhibition Efficiency for 1, 2, and 3 wt% of PE, MEG Inhibited Experiment*

Furthermore, to ascertain their inhibition capacity, the inhibition efficiencies for the systems for the various weight percentages of the inhibitors and that of water and gas only were calculated using the following formulas [33–36].

Inhibition efficiency IE is given as

$$IE = 1 - X \tag{1}$$

Percentage inhibition efficiency is given as

$$\%IE = (1 - X)\% \tag{2}$$

where $X$ is the Inhibition Factor given as

$$X = \Delta P_{inhibited} / \Delta P_{uninhibited} \tag{3}$$

$$\Delta P_{inhibited} = P_i - \Delta P_{inhibited} \tag{4}$$

$$\Delta P_{uninhibited} = P_i - \Delta P_{uninhibited} \tag{5}$$

where

Pi is the initial pressure for both gas and water alone and inhibited systems using PE and MEG. Pinhibited is the final pressure of PE and MEG-inhibited systems
Puninhibited is the final pressure for the gas and water system only.

Inhibition Efficiency versus weight % for Plant Extract (PE) and Monoethylene glycol (MEG) is shown in Figure 10. At 1 wt%, Plant Extract (PE) had a higher inhibition efficiency when compared to 2 and 3 wt% of the Mono Ethylene Glycol (MEG). This could be attributed to the existence of a driving force that was higher at that percentage than in the other weight percentage for the plant extract [16,51]. This was also noted by [36].

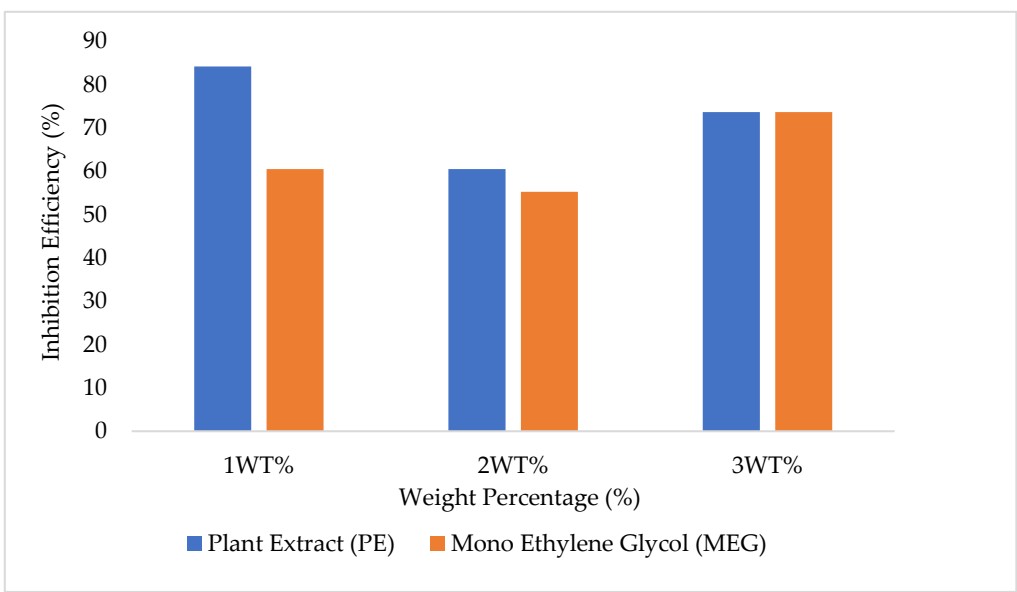

**Figure 10.** Inhibition Efficiency versus Weight percentage for Plant Extract (PE) and Mono Ethylene Glycol (MEG).

The fact that 1 wt% MEG performed better than 2 wt% could be the result of lesser interaction with water molecules at that weight percentage. At 3 wt%, MEG did better than 1 and 2wt% due to more inhibitor content which leads to higher pressure values and lower temperature which aligns with [49].

## 4. Conclusions

The analysis given in the work shows Plant Extract (PE) to be a better inhibitor when compared to Mono Ethylene Glycol (MEG). Based on the recent study, PE was suggested to be eco-friendly and biodegradable and therefore recommended for field trial. This work asserts the position that Plant Extract showed a higher percentage of inhibition in 1 and 2 wt% and had the same efficiency with MEG at 3 wt% [38]. The optimum weight percentage inhibition efficiency is 84.21%. Other weight percentages of PE performed lower than this including all the weight percentages of MEG. The P-T plots for PE favor gas hydrate inhibition better with higher pressure values than those of MEG which was not the case for the uninhibited experiment where the P-T curve tilted heavily to the right giving much room for gas hydrate formation. The various weight percentages of the PE were able to disrupt the thermodynamic equilibrium conditions of the water and gas molecules to lower temperatures and higher pressures which caused a shift in the equilibrium curve to the left allowing for an unstable hydrate formation zone. The 1 wt% shifted the curve to the left better having a pressure of 132 psi at the end of the experiment. The performance at this weight percentage could be attributed to better interaction with the water molecules which did not allow for the formation of gas hydrates. Indeed, 2 wt% and 3 wt% did not perform as well as 1 wt% even though the more the inhibitor content, the higher the temperature drop, and the higher the pressure. This could be probably due to lesser activity or could be a case of over-inhibition leading to a reduction in efficiency.

Based on the Efficiency of Inhibition, it is clearly shown that PE is a better gas hydrate inhibitor than MEG. The Plant Extract (PE) is naturally sourced and environmentally friendly, unlike MEG which is man-made and toxic to both human and aquatic life. It is therefore recommended for field trial.

**Author Contributions:** Conceptualization: J.A.A. and S.S.I.; methodology: V.U.W.-E.; formal analysis, V.U.W.-E. and S.S.I.; writing—original draft preparation, V.U.W.-E.; writing—review and editing, S.S.I. and V.U.W.-E.; supervision, J.A.A. and S.S.I.; project administration, J.A.A. and S.S.I.; funding acquisition, J.A.A., S.S.I. and V.U.W.-E. All authors have read and agreed to the published version of the manuscript.

**Funding:** This research was funded by the TERTIARY EDUCATION TRUST FUND (TETFUND), Institutional Based Research Grant, IBR (2020).

**Institutional Review Board Statement:** Not Applicable.

**Informed Consent Statement:** Not Applicable.

**Data Availability Statement:** Not applicable.

**Acknowledgments:** The Authors are grateful to TETFUND and the Shell JV-Aret Adams Professorial Chair in Petroleum Engineering (SAAPC), University of Port Harcourt.

**Conflicts of Interest:** The authors declare no conflict of interest. The funders had no role in the design of the study; in the collection, analyses, or interpretation of data; in the writing of the manuscript; or in the decision to publish the results.

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
