# Peer review of "The Influence of Plant Extract on the Phase Equilibrium of Structure I Gas Hydrate in a Simulated Offshore Environment"

_2673-7264, doi:10.3390/thermo3010002_

Round 1
Reviewer 1 Report (Previous Reviewer 2)
The authors have addressed my comment and added sufficient detail to make the work more meaningful for the reader. I can now recommend this manuscript for publication.
Minor editorial changes in the English usage can be made at the time of publication
Author Response
Please see attachment

Reviewer 2 Report (New Reviewer)
The article presents an approach to prevent the formation of gas hydrates using inhibitors such as Plant Extract.
The authors often repeat the same information in the introduction e. g. lines 34 and 35 describe the same meaning as in lines 31-33. The authors should correct the introduction.
In lines 193-196, the authors give the main active components of the plant extract. However, the plant extract is quite multi-component. How constant is the plant extract composition depending on the batch of raw materials? How exactly was this plant extract obtained?
The authors said 'Furthermore, the performance of the Plant Extract (PE) is based on the performance of Mono Ethylene Glycol (MEG)"(in line 241). What did the authors mean? MEG was not mentioned in the extract.
Authors should indicate the hydrate formation zone in all relevant figures.
In figures 4, 5, and 6, there are no ticks on the axes. It should be corrected.
The authors conclude that a 1% inhibitor is optimal, and its increase does not affect the effectiveness. However, the content of the inhibitor less than 1% was not considered by the authors. How could this affect the process of hydrate formation?
Author Response
Please see the attachment

Reviewer 3 Report (New Reviewer)
In this work, the authors investigated the effect of a Plant Extract (PE) as a gas hydrate inhibitor on the phase equilibrium of sI gas hydrate. The authors have deficient background knowledge of gas hydrates. The manuscript is full of grammatical errors, logical problems and mistakes. Some sentences are just incomprehensible. Their hydrate formation experiment is thermodynamically incorrect. I recommend rejection of this paper.
1. In the introduction part, there is no in-depth literature review, no problem statement and research objective. There are too many grammatical mistakes.
2. In page 2, row 47-48: What can cause equipment damage if the plugs clump and travel at high velocities? This sentence has no subject.
3. In page 2, row 55: There is a logical problem. Not “gas hydrate formation pressure and temperature” but “gas hydrate formation” can take place during oil and gas production.
4. In page 2 row 66: Spelling mistake. “any” should be “many”.
5. In page 3 row 84: The term “hydrate curve” is ambiguous, it should be “hydrate phase boundary”.
6. In page 3 row 88: The authors should have a uniform reference format.
7. In page 3 row 94: It is not “the mitigation of gas hydrate” but “the mitigation of gas hydrate formation”.
8. In page 3 row 110: The sentence starting with “Prolonged induction time” is not a good sentence. I cannot understand the meaning.
9. In page 3 row 111: not “up take” but “uptake”.
10. In page 3 row 114: MEG and MeOH should be compared as gas hydrate inhibitors, not “gas hydrate inhibition”.
11. In page 4 row 128: Authors do not need to write the complete title of the reference down. Just a reference footnote is enough.
12. In page 4 row 129: “KHI was compared to methanol a commercial THI”. KHIs and THIs has distinct different inhibiting mechanism. Are they comparable?
13. In page 4 row 135-137: The sentence starting with “Varying weight percent…” has incorrect grammar.
14. In page 5 row 182: Why is the combination of rapid temperature increase and excessive pressure decrease an indication of gas hydrate formation? The authors should explain that gas hydrate formation is an exothermic process that consumes guest gas.
15. In page 6 row 190: which kind of water was used? Purity?
16. In page 7 row 237: This flow chart is not well-organized.
17. In page 7 row 243: According to the title of this manuscript, PE should be a THI influencing the phase equilibrium of gas hydrate. Why did authors investigate concentrations as low as 1 wt%, 2 wt% and 3 wt%? THIs usually have much large dosage.
18. Figure 3 has an obvious mistake. Authors conducted a hydrate formation experiment at starting condition of 30 ℃ and 150 psi (approximately equal to 1 MPa). According to the phase diagram of methane hydrate, this P-T condition is outside of the hydrate stable zone.
Author Response
Please see the attachment

This manuscript is a resubmission of an earlier submission. The following is a list of the peer review reports and author responses from that submission.
Round 1
Reviewer 1 Report
The authors used flow loop equipment to experimentally evaluate the Plant Extract (PE) and its effect on the hydrate phase boundary (phase equilibrium). They also compared it with MEG. Although the manuscript is very well written, there is a main concern regarding their approach to evaluate PE and MEG (see below comments). Therefore, the authors need to address the following comments and revised the manuscript with the major revision.
11. Line 41 and 42, Do not put values for temperature and pressures for hydrate conditions, the hydrate can also be below 0°C and 200psi.
22. Line 49, 50: you include propane twice for sII former. However, it is not true that in the oil and gas industry, we have a simple structure II. In reality, different hydrate structures or phases (e.g., 3 or 5 or even more) can form depending on the condition. You can find this in the literature such as
“Experimental measurement of multiple hydrate structure formation in binary and ternary natural gas analogue systems by isochoric equilibrium methods. Energy & Fuels, 35(11), pp.9341-9348.”
33. As you sued the natural gas system in your test, it can form different hydrate structures (see above paper). So, you cannot say structure I in the title. Remove it.
44. Line 50, sH does need to have both heavier components (e.g., cyclo-alkanes) and smaller gases (e.g., methane).
55. Line 72, although the main mechanism of KHIs is to prevent hydrate nucleation and slow down the growth rate, KHIs can also dissociate the hydrate inside the hydrate stability zone depending on the condition as reported in various works such as:
“Anomalous KHI-Induced dissociation of gas hydrates inside the hydrate stability zone: Experimental observations & potential mechanisms. Journal of Petroleum Science and Engineering, 178, pp.1044-1050.”
66. Line 185: Include the composition of natural gas
77. Line 231 to 233, how do you say some pressure drop is due to nucleation and some is due to the hydrate growth. You cannot say it at all by your data.
88. Include the hydrate phase boundary of your gas composition. Because your natural gas is lean gas (94.5% methane), I am afraid if your operating condition in figure 3 is inside the hydrate stability zone.
99. Section 3.1: your approach to evaluate hydrate equilibrium pressure temperature is totally wrong. You need to measure the equilibrium point by step heating after hydrate formation. You cannot compare the PE with MEG by the amount of pressure drop in the system. MEG can only shift the hydrate phase boundary.
Therefore, my main concern is why you used the amount of pressure drop as an index to compare the THIs. Even for LDHI, we do not normally use the amount of pressure drop to evaluate them. The authors need to explain very clearly about this.
110. Section 3.1 it is not clear how much the PE (you said it acts as THI) can shift the hydrate phase boundary. You need to clearly show it to compare it with other THIs.
111. Is there any reason that you used a low dosage of MEG and PE (e.g., 3%)? Normally, THIs are used at a high amount (e.g., 30%) in order to prevent hydrate formation.
112. Section 4 (conclusion): again, I am afraid if you can conclude that “PE is a better gas hydrate inhibitor than MEG”. Because the mechanism of MEG as a THI is shifting hydrate stability zone, so you cannot evaluate them by the amount of pressure drop.
Author Response
The comments raised by the reviewer has been addressed to the best of my ability.

Reviewer 2 Report
This is an interesting study and the use of naturally available materials for clathrate hydrate inhibition is certainly a subject worthy of pursuing. However, there are problems in the manuscript which prevent me from being able to recommend its publication, despite the significant effort taken by the authors on the experiment.
A main problem with the manuscript is the vague description of “plant extract” (PE) used to study the inhibition. The authors give very specific compositions of the constituents of the extract, without mentioning what plant the extract was taken from and how it was prepared? Was there a solvent involved in the extraction? Was any preprocessing of the plant material needed? Is the extract water soluble? Is the extract stable in the entire temperature range needed to run the experiment? How do the authors guarantee uniform composition between different PE preparations?
In the Introduction, the authors mention that monoethylene glycol (MEG) is not a good inhibitor and methanol is usually used. Why do the authors then use MEG to compare with their plant extract?
The text in Figure 2 is so compressed that reading it is not easier than reading the paragraph of text which describes the procedure.
The authors should specify that the temperature – pressure conditions they use are in the natural gas clathrate hydrate formation zone.
It is not clear what Fig. 4 is trying to show. From what I see, the gas pressure decreases more quickly for the systems with PE than the control. The graph is difficult to read with all the colors and data sets plotted in the same area and with the same symbols. The behaviours observed should be more in line with what is shown in Fig. 6.
The phase diagram for the uninhibited system shown in Fig. 5 should be compared to the literature to benchmark the accuracy of the experimental work. The current nearly “vertical” curve seems not correct.
Figure 6 does not show an appreciable benefit of the PE over MEG in slowing down the hydrate formation rate at similar concentrations.
The authors do not describe how they ascertain gas – water – hydrate equilibrium in their phase diagram experiments. Also, if the authors are using PE as a KHI, why are the effects of the phase diagrams of interest? The phase diagrams give the action of the substances as thermodynamic inhibitors?
Author Response
The comments raised by the reviewer has been answered to the best of my ability

Reviewer 3 Report
Please write carefully to achieve consistency. For example, parallel, expression specification, etc. The double coordinate order of Figure 3 and Figure 4 should be inconsistent. References 35 seems to be a draft. Also, references should cite newer ones.
Author Response
Comments raised by the reviewer has been answered to the best of my ability

Round 2
Reviewer 1 Report
The authors could not revise the manuscript according to my comment. The manuscript needs a very major revision, especially for the methodology that they used. Based on my comments, there is a big concern regarding the methodology that they used in their experiment. I appreciate that they used the flow loop, however, using the amount of pressure drop, we can’t conclude which hydrate inhibitor is better because the main mechanism of THI is shifting the hydrate phase boundary neither the induction time nor hydrate growth rate and amount of hydrate formation.
Moreover, the authors could not satisfy my questions, for example, I asked them to provide the gas composition, and they only provide the composition of methane (94.44%) and carbon dioxide (1.54%) which sum up to 95.98%. But what is the rest of the composition? I also asked to provide the phase boundary to make sure that at what subcooling temperature they did the test.
Reviewer 2 Report
The authors have addressed some of the comments and the manuscript has been improved. The main point still exists that without any detail of the plant extract (PE), the work is not reproducible in principle. Whenever the authors have the patent and can provide further details of the material and its preparation the manuscript can be reconsidered.
Other technical points remain, but this is the main problem.